# Community deployment of a synthetic pheromone of the sand fly *Lutzomyia longipalpis* co-located with insecticide reduces vector abundance in treated and neighbouring untreated houses: Implications for control of *Leishmania infantum*

**Raquel Gonçalves**[1], **Cristian F. de Souza**[2], **Reila B. Rontani**[2], **Alisson Pereira**[2], **Katie B. Farnes**[1], **Erin E. Gorsich**[1], **Rafaella A. Silva**[3,4], **Reginaldo P. Brazil**[2], **James G. C. Hamilton**[5]\*, **Orin Courtenay**[1]\*

**1** Zeeman Institute and School of Life Sciences, University of Warwick, Coventry, United Kingdom,
**2** Laboratório de Doenças Parasitárias, Instituto Oswaldo Cruz, FIOCRUZ, Rio de Janeiro, Brazil, **3** Núcleo de Medicina Tropical, Universidade de Brasilia, Brasília, Federal District, Brazil, **4** Ministério da Saúde, Departamento de Imunização e Doenças Transmissíveis, Secretaria de Vigilância em Saúde, Brasília, Federal District, Brazil, **5** Division of Biomedical and Life Sciences, Faculty of Health and Medicine, Lancaster University, Bailrigg, Lancaster, Lancashire, United Kingdom

\* j.g.hamilton@lancaster.ac.uk (JH); orin.courtenay@warwick.ac.uk (OC)

## Abstract

### Background

The rising incidence of visceral leishmaniasis due to *Leishmania infantum* requires novel methods to control transmission by the sand fly vector. Indoor residual spraying of insecticide (IRS) against these largely exophilic / exophagic vectors may not be the most effective method. A synthetic copy of the male sex-aggregation pheromone of the key vector species *Lutzomyia longipalpis* in the Americas, was co-located with residual pyrethroid insecticide, and tested for its effects on vector abundance, hence potential transmission, in a Brazilian community study.

### Methods

Houses within eight defined semi-urban blocks in an endemic municipality in Brazil were randomised to synthetic pheromone + insecticide or to placebo treatments. A similar number of houses located >100m from each block were placebo treated and considered as "True Controls" (thus, analysed as three trial arms). Insecticide was sprayed on a 2.6m² surface area of the property boundary or outbuilding wall, co-located within one metre of 50mg synthetic pheromone in controlled-release dispensers. Vector numbers captured in nearby CDC light traps were recorded at monthly intervals over 3 months post intervention. Recruited sentinel houses under True Control and pheromone + insecticide treatments

**Data Availability Statement:** The data supporting the conclusions of this article are included within the article. The raw data can be accessed at the following URL: https://wrap.warwick.ac.uk/147698.

**Funding:** The work was conducted with the continued supported of the Wellcome Trust https://wellcome.ac.uk Strategic Translation Award (WT091689MF) to JGCH and OC. The funding body played no role in the design of the study, the collection, analysis, interpretation of the data, or in writing the manuscript or decision to submit the paper for publication.

**Competing interests:** The authors have declared that no competing interests exist.

were similarly monitored at 7–9 day intervals. The intervention effects were estimated by mixed effects negative binomial models compared to the True Control group.

## Results

Dose-response field assays using 50mg of the synthetic pheromone captured a mean 4.8 (95% C.L.: 3.91, 5.80) to 6.3 (95% C.L.: 3.24, 12.11) times more vectors (female *Lu. longipalpis*) than using 10mg of synthetic pheromone. The intervention reduced household female vector abundance by 59% (C.L.: 48.7, 66.7%) (IRR = 0.41) estimated by the cross-sectional community study, and by 70% (C.L.: 56.7%, 78.8%) estimated by the longitudinal sentinel study. Similar reductions in male *Lu. longipalpis* were observed. Beneficial spill-over intervention effects were also observed at nearby untreated households with a mean reduction of 24% (95% C.L.: 0.050%, 39.8%) in female vectors. The spill-over effect in untreated houses was 44% (95% C.L.: 29.7%, 56.1%) as effective as the intervention in pheromone-treated houses. Ownership of chickens increased the intervention effects in both treated and untreated houses, attributed to the suspected synergistic attraction of the synthetic pheromone and chicken kairomones. The variation in IRR between study blocks was not associated with inter-household distances, household densities, or coverage (proportion of total households treated).

## Conclusions

The study confirms the entomological efficacy of the lure-and-kill method to reduce the abundance of this important sand fly vector in treated and untreated homesteads. The outcomes were achieved by low coverage and using only 1–2% of the quantity of insecticide as normally required for IRS, indicating the potential cost-effectiveness of this method. Implications for programmatic deployment of this vector control method are discussed.

### Author summary

The predominant sand fly vector of the intracellular parasite *Leishmania infantum*, that causes human and canine visceral leishmaniasis in the Americas, is *Lutzomyia longipalpis*. Vector control tools to reduce transmission are needed. A sex-aggregation pheromone released by male *Lu. longipalpis* attracts female conspecifics which facilitates blood-feeding and transmission. This study, conducted in Brazil, quantifies the effects of community deployment of a synthetic version of the sex-aggregation pheromone, in a controlled-release dispenser, and co-located with lethal insecticide applied to a small area of the household compound or outbuilding wall. 50mg synthetic pheromone dispensers were used since they attracted substantially more vectors than 10mg dispensers. Deploying this novel lure-and-kill method to houses in eight replicate study blocks in two suburban endemic areas, demonstrated that it reduces vector numbers at both pheromone + insecticide treated houses, and neighbouring untreated houses. The presence of chickens (a known additional attraction to blood-seeking female *Lu. longipalpis*) increased the beneficial effects of the intervention. The method used only 1–2% of the quantity of insecticide necessary for IRS for an average sized house. The results demonstrate the efficacy and potential cost-effectiveness of this novel lure-and-kill control method.

## Introduction

Development of effective and sustainable control tools to combat vector-borne transmission of infectious pathogens to humans and animals is a priority, not least for the Neglected Tropical Diseases [1]. *Leishmania* (Kinetoplastida: Trypanosomatidae) are one such group of infectious agents spread by sand flies (Diptera: Psychodidae), and that cause Leishmaniasis presented as a spectrum of clinical and subclinical conditions in humans and animal hosts. *L. infantum* occurs in the Mediterranean countries, parts of central and eastern Asia, and the Americas, where infection variably results in clinical visceral leishmaniasis (VL) in humans and dogs. VL is usually fatal if not treated, and there is no human vaccine.

In the Americas, >95% of the 11,000 reported human VL cases between 2015–2017 occurred in Brazil [2,3] where the principal sand fly vector is *Lutzomyia longipalpis*, and dogs are the epidemiologically important reservoir for human infection [4]. Indoor Residual Spraying (IRS) of insecticides has been the mainstay of vector control since the 1950s, initially against malaria vectors, and later adopted to combat sand fly vectors [5]. The integrated VL control programme currently includes culling dogs that prove positive for anti-*Leishmania* antibody, and human diagnosis and chemotherapeutic treatment [6]. Routine community-based IRS campaigns against sand flies largely have been discontinued or conducted only in response to detection of an incident human VL case [7]. Dog culling is not considered effective, and it is not popular amongst dog owners. Novel and/or complimentary vector control methods are therefore needed, particularly since the current control efforts have not reduced the human VL burden, and transmission is expanding into suburban/urban locations, and into new, and cross-border, regions [3,8,9].

*Lu. longipalpis* sand flies are not particularly endophagic/endophilic: they do not enter well-constructed houses, they are catholic in feeding preference, and are trapped in greatest abundance in animal shelters, and which are often located near to the sleeping locations of the reservoir dogs [10–15]. Reductions in canine and human infection incidence resulting from targeting outdoor aggregations of *Lu. longipalpis* [16–18], suggests, in line with other studies of *Lu. longipalpis* distributions e.g. [10,19–23], that transmission occurs predominantly exterior to houses. Thus, complimentary vector control methods that target vector populations outdoors e.g.[24] may be more effective.

In this respect, there is growing evidence that insecticide-impregnated collars fitted to dogs reduces household *Lu. longipalpis* abundance and *L. infantum* infection prevalence in the vector[25,26]. And that by protecting the reservoir, human infection incidence and clinical VL burdens are also reduced [16,17]. One drawback of this approach is that collar losses from dogs are usually high, making it difficult to maintain individual or community coverage, coupled with the high unit purchase price / replacement cost which is prohibitive for most endemic households [16].

An alternative or complimentary option to reduce transmission relies on exploiting the vector's behaviour in response to volatile pheromones. The sex-aggregation pheromone released by male *Lu. longipalpis* regulates conspecific recruitment to mating leks on or near animal hosts, which facilitates blood acquisition by the female vector, resulting in *L. infantum* transmission [27–29]. Following the recent development of a synthetic copy of this pheromone [30], field experiments demonstrate that it attracts conspecific sand flies to experimental chicken boxes [31], and when co-located with insecticide, takes on a vector lure-and-kill effective mode of action [32]. A recent cluster randomised trial (CRT) of the lure-and-kill method in S.W. Brazil, to test the efficacy of 10mg of synthetic pheromone inserted into controlled-release dispensers, and co-located with residual pyrethroid insecticide, reduced not only the abundance of *Lu. longipalpis* at homesteads by 49%, but also reduced the incidence of

confirmed *Leishmania* infection and clinical *Leishmania* parasite loads in the canine reservoir by 52–53% [18]. These outcomes were not dissimilar to those attributed to the insecticide-impregnated collars that were tested in a parallel trial arm. The collective results of those studies indicate the lure-and-kill method has potential as a public and veterinary health control tool.

In the CRT, the synthetic pheromone was placed at, or near to, chicken roosting sites under the prevailing assumption that the sex-aggregation pheromone worked in synergism with host kairomones to attract the vector [33]. Thus households were recruited in that trial on the basis that they owned both chickens, as the common animal host, and dog(s) in which to measure changes in infection incidence.

In pursuit of developing this vector control method as a public health tool, such a campaign would aim for community-wide coverage including households with and without animal hosts, rather than selective coverage. In VL endemic regions, particularly in urbanised communities, a substantial proportion of households do not keep chickens or any other domestic animals. Hence the efficacy of the lure-and-kill approach in the absence of non-human hosts needed to be determined. Furthermore, community-wide interventions do not achieve 100% coverage, so in this context, there was a need to understand also the potential spatial effects of the synthetic pheromone intervention i.e. if the beneficial effects would spill-over from treated to neighbouring untreated households. Experimental field studies demonstrate that the attraction plume of the synthetic pheromone lure to *Lu. longipalpis* is at least 30m [34].

To address these research questions, a community-based study was conducted in Brazil with the principal aims (i) to quantify the effects of the synthetic pheromone + insecticide intervention on *Lu. longipalpis* vector abundance; (ii) to assess the potential spatial effects of the intervention in untreated houses in the vicinity of treated houses; and (iii) to examine the influence that non-human host ownership could have on the intervention outcomes. In so doing, the study also (iv) estimated the attraction of the vector to different quantities of the synthetic pheromone.

## Materials & methods

### Ethical statement

The research study protocols followed the guidance of Centro de Controle de Zoonoses (CCZ), and were approved by the Secretary of Health, Governador Valadares. No clinical samples or personalized data were collected. The insecticide was supplied by the Brazilian Ministry of Health. Fully informed written signed consent was obtained from all homeowners to participate in the study.

### Study site

The study was conducted in Jardim do Trevo (-18.838581, -41.991113) and Santa Rita (-18.900469, -41.982358) both semi-urban districts of Governador Valadares (GV), Minas Gerais state, Brazil, between October 2018 and January 2020. The population sizes in the two study districts were approximately 4,866 and 19,687 inhabitants respectively (http://populacao.net.br/qual-e-maior-jardim-do-trevo_ou_santa-rita_em-governador-valadares_mg.html accessed 30/8/20).

In Minas Gerais State, 2,456 human VL cases were reported between 2010–2015, with a case fatality of 9.6%; 212 of these cases were reported in GV with case fatality estimates of 8.9%-16.3% [35,36]. The 2008–2015 cumulative VL incidence was 7 cases per 100,000 [37], which is similar to 6.4 per 100,000 just prior to this study in 2017 [38]. The canine infection prevalence in Jardim do Trevo and Santa Rita study districts sampled in 2008–2012 were 27%

(284/1062) and 22% (674/3037) respectively, which were similar to 29% (8,622/29,724) reported for the wider GV canine population [39].

In the study districts, houses are generally small, constructed of concrete with mud or cement floors and tiled roof, with a small garden or outside compound. A large proportion (72.4%) of households own at least one domestic animal; dogs and chickens are the most common and abundant, but recorded in only 47% and 20% of households, respectively. The climate in the region is classified as tropical sub-warm and sub-dry, subject to hot and humid climate throughout the year, with an average temperature of 24.2°C (hottest from November to February) and average annual rainfall of 1,109mm (highest between December and May) (https://en.climate-data.org/location/2879/ accessed 30/08/20).

### Study design

The general design of the intervention study was to recruit houses within eight defined geographical study blocks, within which houses were then randomly assigned to one of two treatments, either synthetic pheromone co-located with insecticide ("Pheromone" arm 1), or placebo ("Untreated" arm 2). The within-block Untreated houses represented the group in which potential spill-over intervention effects on vector numbers from Pheromone houses was measured. A third group of houses peripheral to each study block boundaries were recruited and assigned to the placebo treatment to act as the "True Control" group (arm 3). Thus, the trial comprised three arms, Placebo and Pheromone groups within study blocks, and the True Control group exterior to each study block. The effect of the synthetic pheromone co-located with insecticide on the numbers of *Lu. longipalpis* in Pheromone and Untreated arms were quantified relative to numbers in the True Control arm, measured during 3 months post-intervention.

### Recruitment

The districts of Jardim do Trevo and Santa Rita were indicated by the local health authorities as suitable sites for study given evidence of chronic *L. infantum* transmission as described above. The two districts were located approximately 4.75 km apart. Four blocks (approx. 30,000m² per block) of contiguous houses in each district were delineated on the criteria that each block was spatially separated from its nearest neighbouring block by >100m (8 blocks in total) (Fig 1). Post selection, the mean nearest neighbour separation distances between study blocks was 229 metres (SD: 39.3m, range: 198-279m) and 817m (SD: 223.5m, range: 704-1152m) in Jardim do Trevo and Santa Rita districts, respectively.

Within study blocks, residential houses were eligible for recruitment following the criteria that (i) the householders were normally resident in the house; (ii) the accommodation had a private outside compound or garden to locate the intervention i.e. was not an apartment block; (iii) the residents were at home between the hours of 07:00–09:00 hrs and 17:00–19:00 hrs to allow access for sand fly trap setting and collection; and (iv) the householders confirmed that IRS and/or other chemical treatment had not been applied to the property within 12 months prior to the start of the study (Fig 2). The nearest neighbour median distance of recruited houses within blocks was 16m (IQR: 12.7–20.9m; range: 2.2–45.2m).

In addition to houses recruited within study blocks, True Control houses recruited outside each of the eight study blocks followed the same selection criteria, whilst ensuring that the minimum distance of each house to the associated study block boundary was >100m. This was in order to exclude potential contamination from the ~30m attraction plum of the synthetic pheromone, or from naturally dispersing *Lu. longipalpis* (usually <100m [28,40–43]). The aim was to recruit a similar number of True Control houses as Untreated houses per study

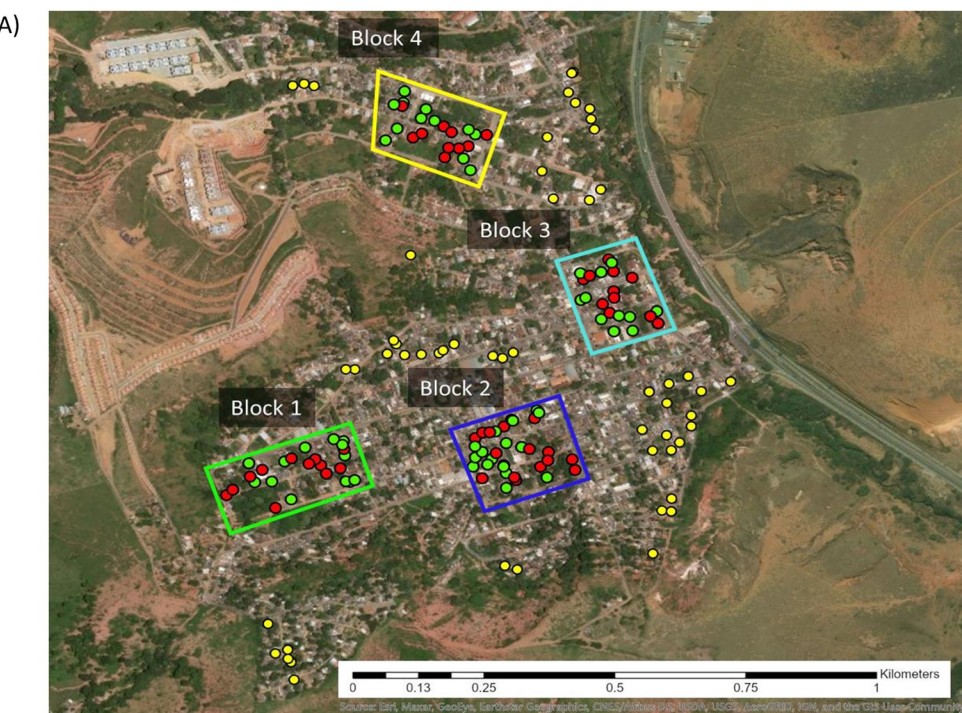

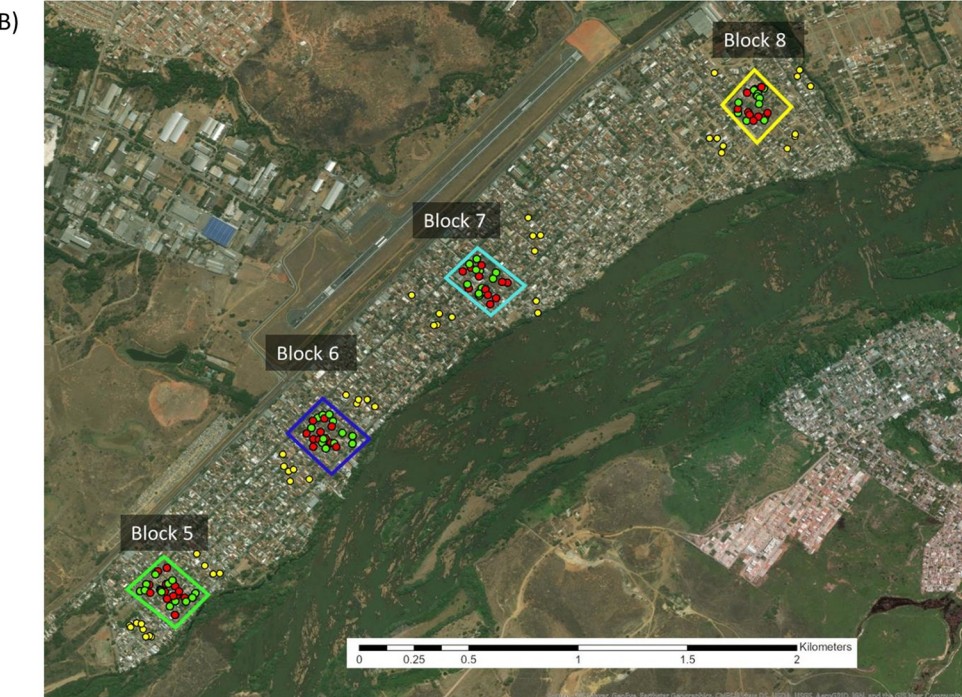

**Fig 1. Locations of the recruited blocks of houses in the two suburban study districts in Governador Valadares, Minas Gerais state, Brazil: Jardim do Trevo (A) and Santa Rita (B).** Houses are colour coded by intervention arm allocation: True Control (yellow), Untreated (green), and Pheromone (red). The map was created in ArcGIS Pro software by Esri using the 'WorldImagery' basemap (Esri. basemap. "World Imagery Map" 28 Sept 2019. https://www.arcgis.com/home/item.html?id=c1c2090ed8594e0193194b750d0d5f83 (accessed 5/11/20).

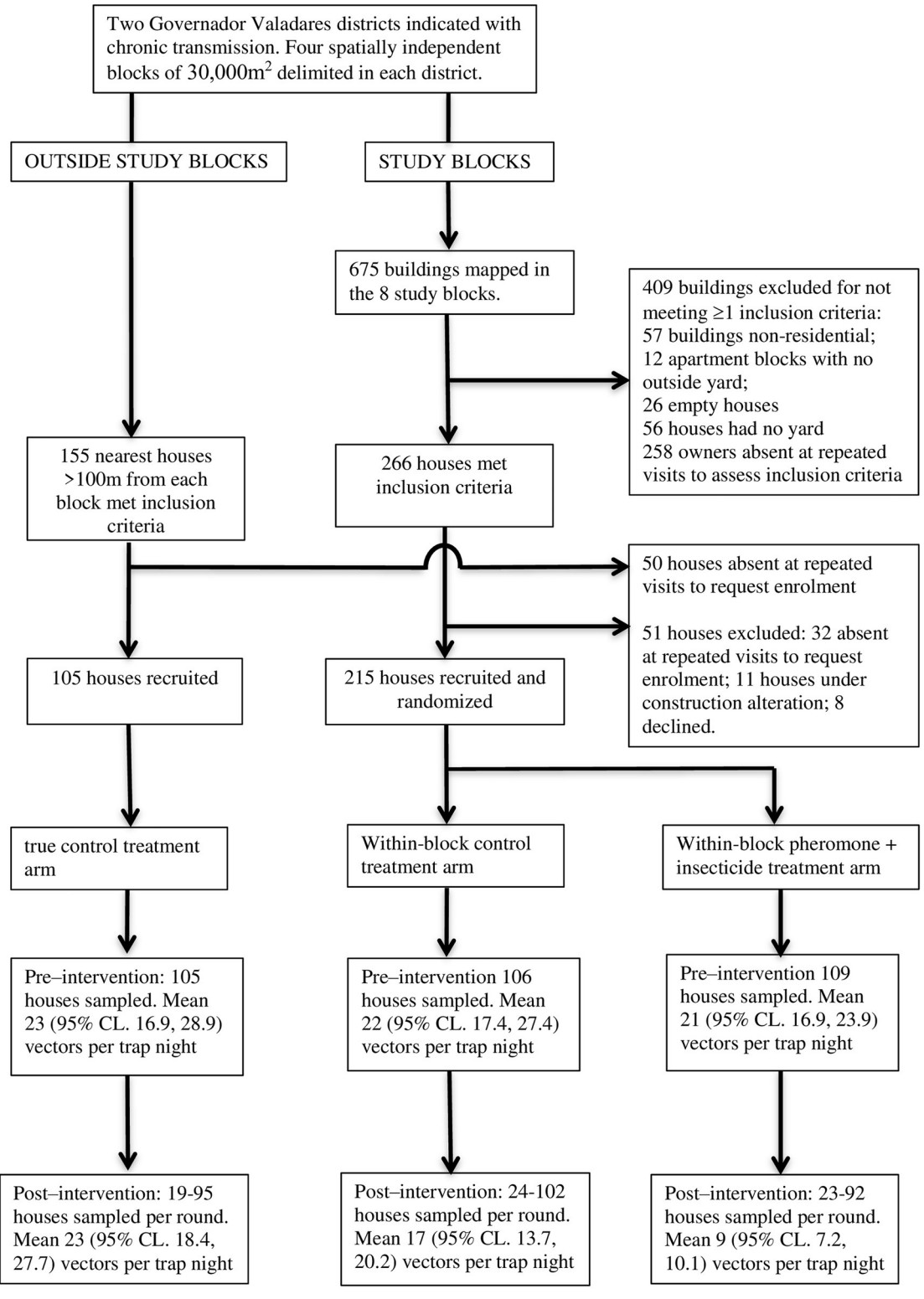

**Fig 2. Study design and structure.**

block (Fig 2). The mean distance of recruited True Control houses to each block boundary was 159m (SD 29.2m, range: 120m-197m).

## Treatment randomisation

Recruited houses within study blocks were listed in order of recruitment visit date, and a random number generated in STATA was assigned to each house. Houses were then ordered from smallest to highest random number, and the first half assigned to placebo treatment, and the second half to synthetic pheromone + insecticide treatment. The procedure was repeated for each of the eight study blocks in turn. Pheromone houses were the nearest neighbour to 52.1% of the Untreated houses, and to 47.9% of other Pheromone houses, indicating the success of the within-block treatment randomisation process. All True Control houses were placebo treated.

## Interventions

**Synthetic pheromone + insecticide treatment (arm 1).** Houses within study blocks which were assigned to the Pheromone arm were treated with the synthetic pheromone (±-9-methylgermacrene-B [CAS RN: 183158-38-5]) [30] which is the racemate copy of the (*S*)-9-methylgermacrene-B pheromone produced by male *Lu. longipalpis* in the study region [44]. 10mg of synthetic pheromone was sealed in 8 cm × 3 cm polythene sachet dispensers designed for controlled release (Russell-IPM Ltd. UK). Five × 10mg dispensers were zip-tied together and located outside treated households suspended 1.5m from the ground by a wire cable fixed to a tree or to a secure pole, aiming to be within 1m of the insecticide treated location as described below.

Henceforth, we refer to the five × 10mg dispensers as 50mg of synthetic pheromone for convenience but stress the technical difference: the amount of pheromone released per unit time from a dispenser loaded with 50mg would be smaller than that released by five individual 10mg dispensers.

Alphacypermethrin SC (Alfatek 200 SC 20%, Rogama; lot number 007/18; expiry August 2020), was sprayed to cover a $2.6m^2$ surface area of a boundary wall or outbuilding exterior wall, selected to be furthest (usually 4-5m) distance from the house entrance. Where this was not possible, the insecticide was applied to an animal shed (in 6% of all houses, or 7.5% of pheromone + insecticide treated houses). The Brazilian Ministry of Health recommends alphacypermethrin SC be applied at a delivery concentration of 40mg active ingredient (a.i.)/$m^2$ for purposes of IRS [6]. However, prior quality control studies of IRS using alphacypermethrin SC applied to adobe wall surfaces inside Bolivian houses, reported that 90% of 57 treated houses, and 84% of 480 filter papers for insecticide capture, received less than the target delivery concentration [45]. Similarly, quality control of IRS using DDT in India, showed that 87% of 560 houses received concentrations under the target dose of 1g a.i./$m^2$ [46]. Thus, in the present study, to compensate for possible suboptimal insecticide delivery, the insecticide was prepared at the equivalent of 80mg a.i./$m^2$ aiming for a target delivery concentration of 40mg a.i./$m^2$ [6]. The insecticide was applied using an 8L Guarany knapsack sprayer (Guarany Ind. Com. Ltda., Itu, Sao Paulo, Brazil), which produces a spray swatch of 75 cm width, therefore the wall surface area was delimited to 1.45m wide × 1.80m high, to achieve a treated surface of approximately $2.6m^2$ per house. Similar calculations were made if instead animal shelters were sprayed. Quality control samples from the spray tank solution, and sprayed walls were systematically collected to quantify the delivered and residual a.i. concentrations. There were no reports of IRS and/or other chemical treatments within the 12 months prior to the intervention, or during the study follow-up period.

**Table 1. The number of households entomologically sampled per trial arm in each sample round pre- and post-intervention.**

| Sample round | True Control arm[1] | Untreated arm | Pheromone arm | Total | Mean days from intervention (95% C.L.s) |
|---|---|---|---|---|---|
| 1 Pre-intervention | 105 | 106 | 109 | 320 | -107 (-111.0, -103.6) |
| Post-intervention | | | | | |
| 2 | 95 | 102 | 92 | 289 | 31 (30.2, 31.0) |
| 3 | 93 | 102 | 87 | 282 | 61 (60.0, 61.5) |
| 4 | 19 | 24 | 23 | 66 | 94 (92.3, 95.8) |
| Total trap nights | 312 | 334 | 311 | 957 | |

[1] True Control and Untreated houses both received placebo treatment. Pheromone houses and Untreated houses were located within study blocks

**Within-block Untreated and True Control placebo (arms 2 and 3).** Houses assigned to placebo treatment, i.e. Untreated houses within-blocks, and the True Control houses outside study blocks, were sprayed with sham insecticide (water) and provided with identical pheromone dispensers which contained no pheromone, both located as described above. Different spray tanks were used to spray pheromone-treated and placebo-treated houses to avoid contamination.

## Pre-intervention survey and intervention follow-up sampling regime

Pre-intervention, 105 True Control houses, 106 Untreated houses, and 109 Pheromone houses (total 320 houses), were sampled for sand flies using 50mg of synthetic pheromone (Fig 2; Table 1). Blocks were sampled in randomized block order.

The intervention was applied to each house on a single occasion between 30/5/19 and 23/9/19, in the same sequential order as the pre-intervention survey for blocks and houses. Follow-up entomological sampling was conducted on a single night at approximately 30, 60 and 90 days post-intervention (Table 1). Between pre-intervention recruitment, treatment, and first follow-up sample, 10, 4 and 17 houses in each arm were lost-to-follow-up (Table 1 and Fig 2). The numbers of houses sampled post-intervention varied between followed-up rounds: 289 houses 30 days post-intervention (across all 8 blocks); 282 houses at day 60 (across all 8 blocks); and 66 houses at day 90 (3 blocks) (S1 Table). The 90 day sample was limited to houses in only three blocks (blocks 1, 5 and 6) due to a late decision to measure the duration of entomological impact for an additional month; the sample thus included all houses that had not had the pheromone dispensers removed after the 60 day follow-up sample. Bias in effect estimates due to subsampling only these three blocks was not detected.

## Sentinel houses

To monitor changes in vector abundance at a finer temporal scale, a subset of 40 sentinel houses (17 True Control houses and 23 Pheromone houses) were recruited across the 8 study blocks selected on showing high numbers of *Lu. longipalpis* by the pre-intervention survey. Sand fly trapping was conducted from 6 to 112 days post intervention at a median interval of 7 days (range: 7–13 days); individual households were sampled on 2–9 (True Control arm) and 6–9 (Pheromone arm) occasions each (S2 Table).

## Synthetic pheromone concentration attraction response

Prior to the intervention study, the attraction response of *Lu. longipalpis* to synthetic pheromone concentrations were evaluated by sand fly trapping on two nights per household using a

CDC light trap with pheromone dispenser(s) attached to the underside of the lid, and light bulb removed. A total 295 houses (33–49 houses in each study block) were sampled on a single night during an initial trapping period (04/02/19 to 28/2/19) using 10mg of pheromone. This was followed-up using 50mg of pheromone for one night during a second trapping period (1/4/19 to 29/4/19) (S3 Table). The mean interval between trap nights was 53 days (95% C.L.: 51.7, 53.6; range: 21–91 days). Blocks were initially sampled in randomized block order, and houses sampled in the same order at both time points.

Due to the known seasonality of *Lu. longipalpis*, and thus potential intrinsic differences in abundance between the two sampling rounds described above, a further comparison of *Lu. longipalpis* dose-response was conducted at an additional 38 households across the 8 study blocks in May 2019. Half of the houses were sampled using a CDC trap fitted with 10mg of synthetic pheromone, the other half fitted with 50mg pheromone. The trapping took a total 4 days (20-23/5/2019) to complete.

### Sand fly trapping

Sand flies were monitored using a miniature CDC miniature light trap, with the light bulb removed, placed at 1.5m height aiming to be within 1m from the insecticide or placebo sprayed wall location. Traps were set and collected before 19:00 hrs and 09:00 hrs, respectively. At Pheromone houses, the existing 50mg pheromone lure was translocated to the CDC trap for the night of monitoring. At Untreated and True Control houses, a 50mg pheromone dispenser, with similar contemporary age to those in Pheromone houses, was fitted to the CDC trap for the night of monitoring only. All pheromone lures were marked with the date that they were first opened and subsequent time in the field. As the CDC light trap power sources varied (6 volt batteries, n = 9; 12 volt batteries, n = 8; non-rechargeable 3 volt batteries, n = 31), the battery type used was recorded. Comparison of the baseline sand fly catch data revealed no significant differences between battery types (LRT $\chi^2_3$ = 3.85, p = 0.146), and was thus omitted from further analyses.

### Sandfly identification

Sand flies were counted, sexed, and preserved in 70% alcohol. In GV, *Lu. longipalpis* accounts for 90–98% of all sand fly species captured by CDC light traps set in peridomestic locations [35,47], and the synthetic pheromone has a very high degree of species specificity[32,47]. Hence, sand flies were not routinely identified during this study. However, males were confirmed according to their genitalia and presence of tergal spots on abdominal tergites, following the identification key of [48].

### House densities and nearest-neighbour distances

All buildings within blocks, and True Control houses, were georeferenced (UTM zone 24) using a Garmin Etrex10 and visualised using ArcGis Pro v2.4 (ESRI 2019). The distances between houses and blocks, and the densities of Pheromone houses within 30m radius of all houses, were calculated using R v4.0.0 (R Core Team 2020) with the distGeo function in the geosphere package [49].

### Demographic and implementation logistic measures

The numbers and types of domestic non-human hosts per household were recorded by observation or by questioning the head of the household. Variables associated with the logistics of the pheromone + insecticide deployment were recorded at the time of trapping, including (i)

the location of the CDC light trap; (ii) the location of the synthetic pheromone dispenser; (iii) the substrate onto which the insecticide was sprayed; and (iv) the distance between the pheromone dispenser and the insecticide treated structure (S4 Table).

## Statistical analyses

The numbers of *Lu. longipalpis* captured per house trap night in Pheromone and Untreated arms were each compared to contemporary catch numbers in the True Control arm. Analyses of the intervention effects were computed using mixed-effects negative binomial models, expressed as incidence risk ratio (IRR) i.e. the ratio of the cumulative incident number of vectors trapped in each treatment arm. Models included interaction terms: arm × pre-/post-intervention, and arm × days from intervention, as appropriate, and random intercepts for study blocks were fitted (study blocks being the higher level of structuring in the data [50]). Effect estimates were calculated using post-estimation LINCOM routines in STATA v.15 (StataCorp LP, College Station, TX). Variables describing host abundance and the implementation logistics measures, were each evaluated for significance by log–likelihood ratio test (LRT) of nested models.

## Results

### Synthetic pheromone concentration attraction responses

Prior to the intervention trial, sand fly trapping was conducted at 295 households to compare the attraction of *Lu. longipalpis* to 10mg versus 50mg of synthetic pheromone fitted to CDC light traps (n = 2 x 295 trap nights). The 50mg lure captured a geometric mean 3.7 (95% C.L.: 3.32, 4.10) female, and 11.3 (95% C.L.: 10.10, 12.65) male *Lu. longipalpis* per house trap night. These compared to 1.6 (95% C.L.: 1.47, 1.79) females and 3.7 (95% C.L.: 3.27, 4.16) males using 10mg pheromone (S1 Fig). Sand fly numbers captured per trap night were male biased and the sexes not highly correlated (10mg: mean M/F ratio 3.7 [C.L.: 3.10, 4.30]; $r^2 = 0.63$; 50mg: mean M/F ratio 4.6 [C.L.: 4.07, 5.09]; $r^2 = 0.74$).

Accounting for the study design, including the variation between the eight study blocks, the mixed effect model predicted that the 50mg lure captured a mean 4.8 (95% C.L.: 3.91, 5.80) times more female *Lu. longipalpis* than the 10mg lure (z = 15.5, p<0.0001). The time interval between household consecutive captures did not modify these estimates (z = 0.39, p = 0.697). The equivalent mean increase of 3.9 (95% C.L.: 3.30, 4.56) in male *Lu. longipalpis* numbers was not dissimilar.

In 88.5% (261/295) of households, the 50mg lure captured greater absolute numbers of *Lu. longipalpis* than the 10mg lure; the geometric mean increase was 11.3 (95% C.L.: 10.0, 12.8) flies (sexes combined) per trap night. A lower number of *Lu. longipalpis* were captured using 50mg vs 10mg lures in 8.1% (24/295) of households (geometric mean 3.6 [95% C.L.: 2.53, 4.99] per trap night); and a zero difference was observed in 3.4% (10/295) of households. In only two households were there zero sand fly captures on both trap nights (S2 Fig).

The 38 additional houses that were sampled within 4 days, half fitted with 10mg of pheromone, the other half fitted with 50mg pheromone, recorded geometric means of 1.9 (95% C.L.: 1.37, 2.51) and 7.2 (95% C.L.: 4.61, 11.19) female, and 3.8 (95% C.L.: 2.44, 5.91) and 19.2 (95% C.L.: 12.61, 29.21) male *Lu. longipalpis*, respectively. This was equivalent to a predicted 6.3 (95% C.L.: 3.24, 12.11) and 5.4 (95% C.L.: 3.22, 8.89) times more female and male *Lu. longipalpis* captured using 50mg lures, respectively (z>5.44, p<0.0001 in each case).

Based on the greater attraction of 50mg of synthetic pheromone, this amount was used for the intervention study, at baseline and for follow-up monitoring.

### Pre-intervention sampling

Vectors were collected pre-intervention in 105, 106, and 109 households which were subsequently assigned to True Control, Untreated, and Pheromone arms, respectively (Table 1 and Fig 2). There were no differences in the pre-intervention mean numbers of female or male *Lu. longipalpis* captured between the three trial arms ($z<1.03$, $p>0.174$); the geometric mean numbers of females captured per household trap night were 4.1 (95% CL: 3.47, 4.86), 3.2 (2.61, 3.83), and 4.0 (3.41,4.78), respectively (median values are displayed in S3 Fig).

### Post-intervention effect estimates

Post intervention, sand fly captures were conducted in three sample rounds, at 289, 282, and 66 houses, respectively, at a mean interval of 30 days (95% C.L.: 29.5, 31.1) (Table 1). The geometric mean numbers of females captured per household over the three month follow-up period were 3.5 (95% CL: 3.0, 4.12), 3.1 (2.74, 3.60), and 2.5 (2.10, 3.00) in True Control, Untreated, and Pheromone arms, respectively (median values are displayed in S3 Fig). Accounting for variables describing the trial design and time (days) under intervention, the cumulative numbers of female *Lu. longipalpis* captured in the Pheromone arm by the end of the follow-up period were reduced by a mean 59% (95% C.L.: 48.7%, 66.7%), compared to in True Control arm; the relative changes in male *Lu. longipalpis* numbers showed a similar pattern (Table 2).

Spill-over effects of the pheromone + insecticide treatment to neighbouring Untreated households (within study blocks) was observed. The vector numbers captured at Untreated houses were reduced by a mean 24% (95% C.L.: 0.050%, 39.8%) as compared to True Controls (Table 2). Compared directly to Pheromone houses (i.e. not to True Controls), the intervention reduced female *Lu. longipalpis* numbers by 44% (95% C.L.: 29.7%, 56.1%) relative to the mean reduction achieved in Pheromone houses (IRR = 0.56 [95% C.L.: 0.439, 0.703], $z = -4.90$, $p<0.001$). A similar pattern was observed for male *Lu. longipalpis* (IRR = 0.50 [95% C.L.: 0.412, 0.611], $z = -6.88$, $p<0.001$).

The duration of the intervention effect against female vector numbers, measured over the three months, lasted for at least 2 months post intervention in both Pheromone and Untreated arms, reaching a 70.7% (95% C.L.: 59.2%, 78.9%) reduction in Pheromone houses within the first month, but which reduced to pre-intervention levels by about 90 days post intervention (Figs 3 and S4).

### Sentinel houses

The temporal pattern was similar to that measured longitudinally in the 40 sentinel households (Figs 4 and S5). About one week post-intervention there was a 59% reduction (IRR = 0.41) in female *Lu. longipalpis* in Pheromone houses compared to True Control houses, reaching 85%

**Table 2. Intervention effect estimates (IRR) on female and male *Lu. longipalpis* numbers across treatment arms compared to the True Controls after 3 months follow-up.**

| Treatment arm[1] (n) | Female IRR (95% C.L.) | P< | Male IRR (95% C.L.) | P< |
|---|---|---|---|---|
| True Control (307) | referent | | referent | |
| Untreated (331) | 0.76 (0.602, 0.950) | 0.016 | 0.79 (0.645, 0.966) | 0.022 |
| Pheromone (296) | 0.41 (0.333, 0.513) | 0.0001 | 0.40 (0.334, 0.490) | 0.0001 |

[1] True Control and Untreated houses both received placebo treatment. Pheromone houses and Untreated houses were located within study blocks

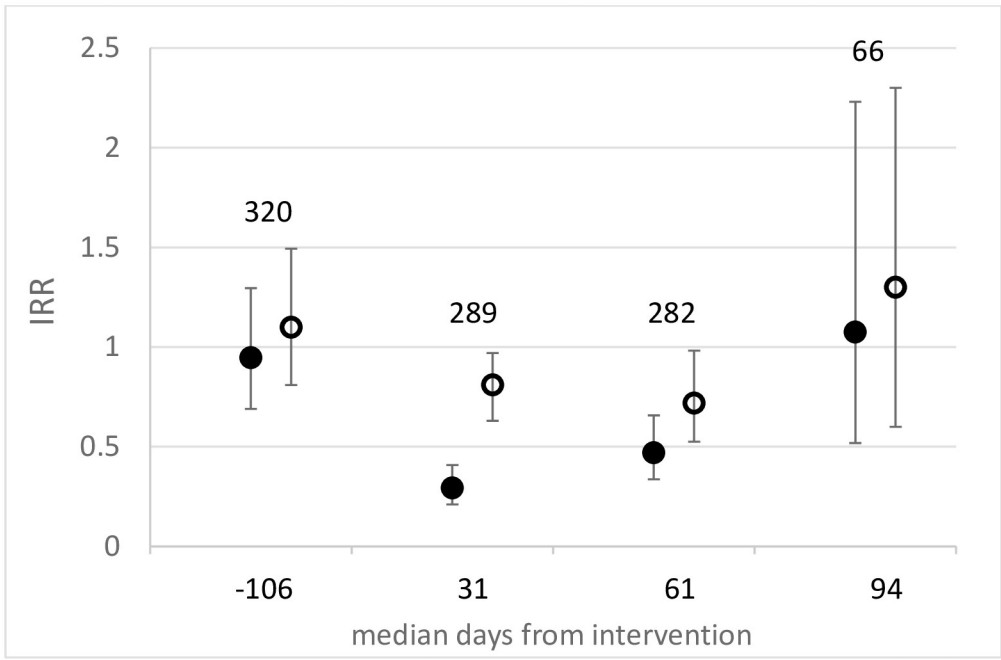

**Fig 3. Intervention effect (IRR) against female *Lu. longipalpis* in Pheromone houses and Untreated houses relative to True Control houses, measured pre-intervention and at approximately monthly sampling rounds post-intervention.** IRR (points), 95% C.L.s (bars), and sample sizes (numbers above bars).

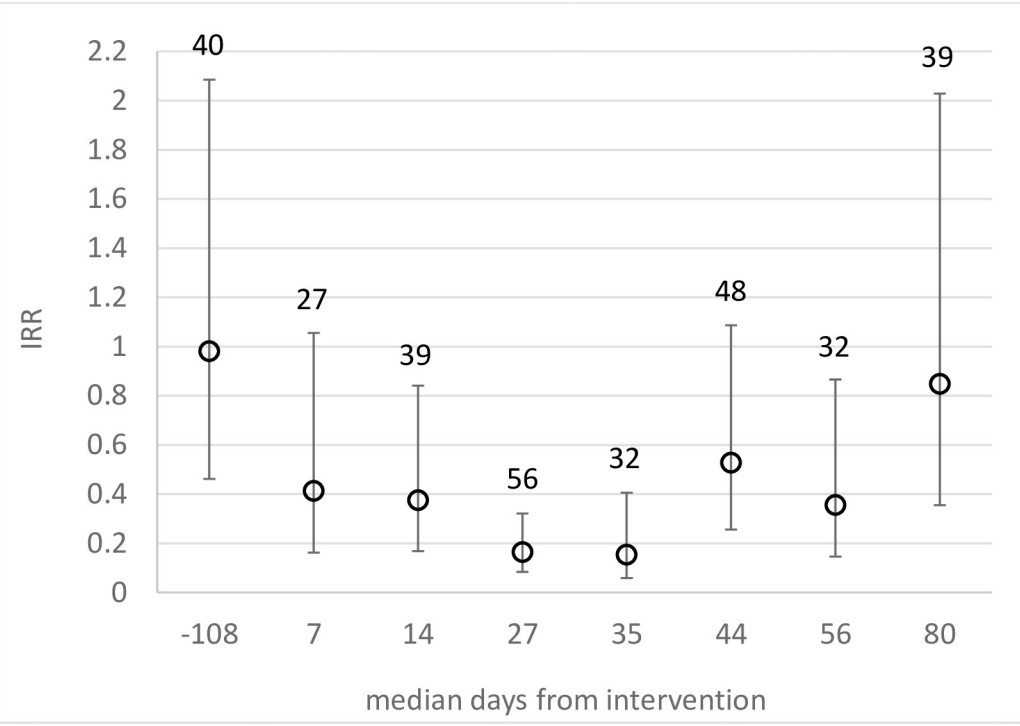

**Fig 4. Intervention effect (IRR) against female *Lu. longipalpis* in sentinel Pheromone houses relative to True Control houses, measured pre-intervention (days<1) and at follow-up post-intervention (days>0) at median interval of 9 days (range: 7–13 days).** IRR (points), 95% C.L.s (bars), and sample sizes (numbers above bars).

**Table 3. Intervention effect estimates (IRR) on female and male *Lu. longipalpis* numbers in Pheromone compared to in True Control sentinel houses over 3 months post-intervention.**

| Treatment arm (n) | Female IRR (95% C.L.) | P< | Male IRR (95% C.L.) | P< |
|---|---|---|---|---|
| True Control (129) | Referent | | referent | |
| Pheromone (185) | 0.30 (0.212, 0.433) | 0.0001 | 0.29 (0.204, 0.399) | 0.0001 |

(IRR = 0.15) by about one month post intervention. Significant reductions (>47%) were observed for at least 2 months post intervention (Fig 4).

The mixed effects model predicted a 69.6% (95% C.L.: 56.7%, 78.8%) cumulative protective response against female, and similarly 71.4% (95% C.L.: 60.0%, 79.6%) against male, *Lu. longipalpis* in sentinel Pheromone households (Table 3).

### Variation in intervention effects between study blocks and districts

There was no detectable differences in intervention effects on female or male *Lu. longipalpis* between the two study districts (LRT $\chi^2_1$ <1.65, p>0.20). The effects appeared broadly similar between the eight blocks, though confidence intervals were expectedly broad (Fig 5). Compared to True Controls, mean values of IRR<1 were observed in 7/8 and 6/8 study blocks for Pheromone and Untreated arms, respectively. IRR values within blocks for the two arms were correlated (pairwise correlation coefficient: $r^2$ = 0.76) suggesting potential inherent differences between the study blocks: consistently strong intervention effects were apparent in blocks 3, 6, and 8; and less strong in blocks 4 and 7 (Fig 5).

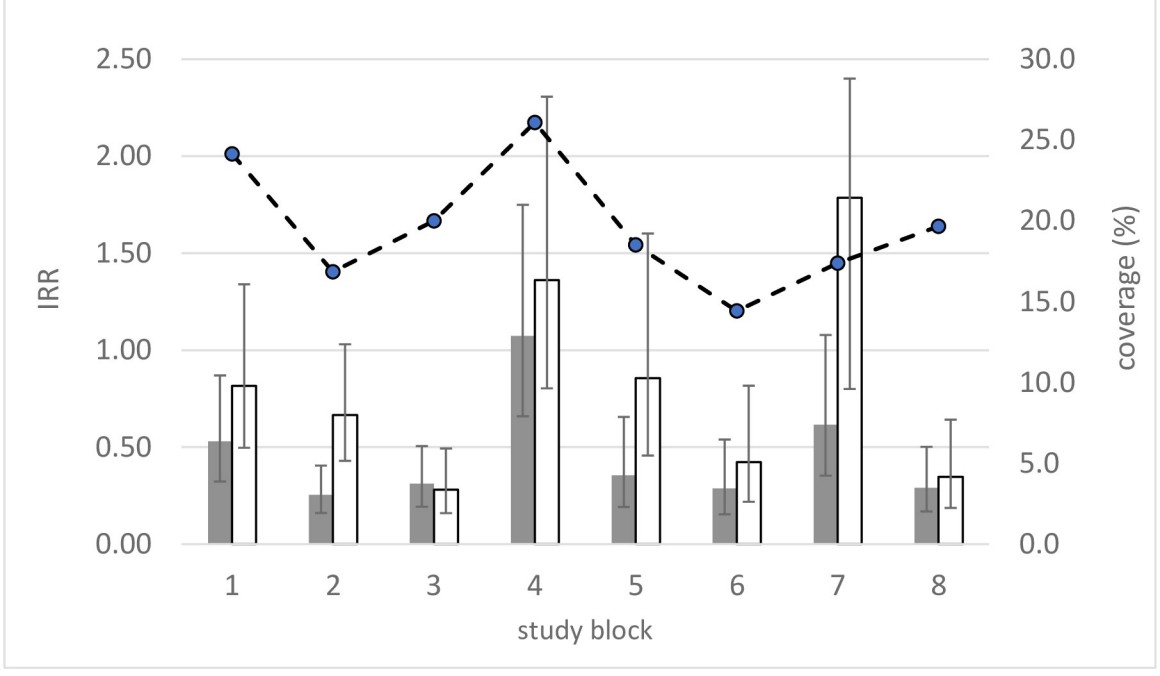

**Fig 5. Variation in intervention effects (IRR) against female *Lu. longipalpis* between study blocks at Untreated houses, and Pheromone houses, both relative to True Controls.** Districts: Jardim do Trevo (blocks 1–4) and Santa Rita (blocks 5–8). IRR (points) and 95% C.L.s (bars).

The variation in IRRs across study blocks was not significantly associated with the percentage of total houses that received the pheromone + insecticide treatment (intervention coverage). The mean coverage was 19.6% (range: 14.4%-26.1% per study block) (Fig 5).

## Geographical proximity to pheromone + insecticide treated houses

The spill-over effect of the intervention from Pheromone houses to Untreated houses warranted further investigation. The variation in geographical proximity of an Untreated house to a Pheromone house did not significantly influence the intervention effects against either female or male *Lu. longipalpis* (LRT $\chi^2_1$ <1.52, p>0.218) (Fig 6). Similar proportions of Untreated (52.1%) and Pheromone (47.9%) houses were nearest neighbour to a Pheromone house, and which were located at a similar nearest neighbour distance (Untreated houses: median 15m [IQR: 11.3–20.2m], range: 2.2–45.2m; Pheromone houses: median 18m [IQR: 13.8–23.0m], range: 3.5–40.9m) (S6 Fig and S4 Table).

Neither were the intervention effects modified by the presence/absence, or the variation in the number, of Pheromone houses within a 30m radius of an Untreated house (LRT $\chi^2_1$

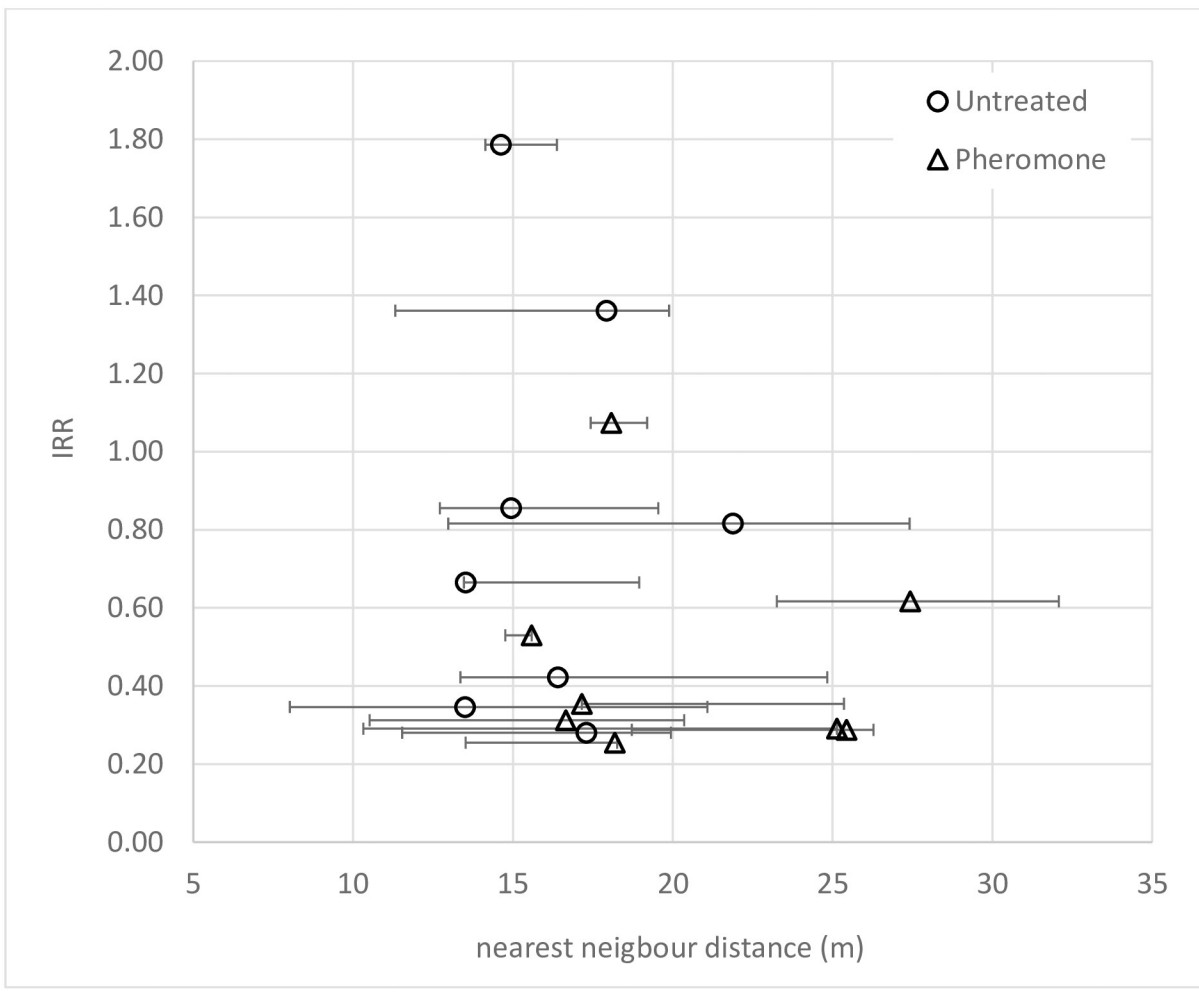

**Fig 6. Relationship between the intervention effect (IRR) and the median distance (meters) to the nearest neighbouring Pheromone house within study blocks.** IRR shown for Pheromone houses (triangles) and Untreated houses (circles). X-axis error bars represent the IQR of the median nearest neighbour house distances. IRR estimates are relative to True Control houses; see Fig 5 for IRR confidence intervals.

>1.50, p>0.473). A similar proportion of Untreated houses and Pheromone houses (98.0% versus 91.1% respectively) had at least one (range: 0–3) Pheromone house within 30m radius ($\chi^2_1$ = 2.18, p = 0.139).

### Influence of non-human hosts on *Lu. longipalpis* numbers and intervention outcomes

Similar proportions of households in the three intervention arms (69.5%, 75.5% and 75.0%) maintained one or more non-human host ($\chi^2_2$ = 1.10, p = 0.578), the most frequent and abundant in biomass being chickens (19.7% of households) and dogs (46.6% of households). The proportions of households that owned chickens were similar between trial arms (19.0%, 14.7%, and 23.9%, respectively) ($\chi^2_2$ = 2.66, p = 0.264).

The pre-intervention geometric mean number of *Lu. longipalpis* captured at households that kept chickens (n = 63) was 19 (95% C.L.: 15.2, 23.9), being significantly higher than 13 (95% C.L.: 11.7, 15.1) captured in households that did not keep chickens (n = 257) (z = 2.91, p = 0.004). The geometric mean number of *Lu. longipalpis* also appeared to positively increase with numbers of chickens per household (LRT $\chi^2_2$ = 9.31, p = 0.009) (Table 4). The presence/absence, or number, of household chickens did not affect the *Lu. longipalpis* M:F sex ratios in CDC trap captures (z<1.06, p>0.288).

In contrast to chicken ownership, there were no significant associations between *Lu. longipalpis* numbers and the ownership or number of dogs or other hosts in a household (z<1.70, p>0.09 in all cases).

The presence of chickens, rather than the numbers of chickens, significantly modified the intervention interaction coefficients, resulting in effect estimates against female *Lu. longipalpis* of 72% (C.L.: 55.3, 82.8%) and 54% (C.L.: 24.9, 71.4) at Pheromone and Untreated households, equivalent of 1.65 and 1.46 times increased reduction in vector abundance, respectively, relative to houses without chickens (Table 5). The intervention did not significantly alter the M:F sex ratio (S7 Fig).

Ownership of chickens or other hosts did not significantly modify the intervention effect estimates for male *Lu. longipalpis*, nor did ownership of animals other than chickens influence

**Table 4. The relationship between the pre-intervention geometric mean numbers of *Lu. longipalpis* captured and the numbers of chickens maintained at households.**

| Numbers of chickens owned | Geometric mean number of *Lu. longipalpis* (95% C.L.) | IRR (95% C.L.) | P< = | N houses |
|---|---|---|---|---|
| 0 | 13 (11.7, 15.1) | referent | | 257 |
| 1–5 | 17 (12.2, 24.3) | 1.40 (1.02, 1.92) | 0.039 | 33 |
| >5 | 21 (15.7, 29.0) | 1.47 (1.06, 2.03) | 0.021 | 30 |

**Table 5. The intervention effects on female *Lu. longipalpis* estimated by models adjusted by the presence/absence of household chickens compared to the unadjusted model.**

| Treatment arm[1] (n) | Unadjusted[2] IRR (95% C.L.) | P< | Adjusted[3] IRR (95% C.L.) | P< |
|---|---|---|---|---|
| True Control (307) | referent | | referent | |
| Untreated (331) | 0.76 (0.602, 0.950) | 0.016 | 0.46 (0.286, 0.751) | 0.022 |
| Pheromone (296) | 0.41 (0.333, 0.513) | 0.0001 | 0.28 (0.172, 0.447) | 0.0001 |

[1] True Control and Untreated houses both received placebo treatment. Pheromone houses and Untreated houses were located within study blocks

[2] The unadjusted mixed effects model excludes the intervention arm × chicken ownership interaction term; the estimates are as shown in Table 2.

[3] The adjusted model includes the significant intervention arm × chicken ownership interaction term.

the intervention outcomes on female *Lu. longipalpis* (intervention arm × host variable interactions: z<1.61, p>0.106 in all cases).

### Implementation measures

Variations from the intended practise of intervention deployment were recorded at the time of trapping, namely the location of the CDC light trap, the location of the synthetic pheromone dispenser, the substrate onto which the insecticide was sprayed, and the distance between the pheromone dispenser and the insecticide treated structure. None of these variables significantly altered the effect estimates (LRT $\chi^2_{(1-3)}$ <3.64, p>0.10) (S5 Table).

## Discussion

This study demonstrated that the synthetic pheromone + insecticide intervention reduced vector abundance by a cumulative 59% (95% C.L.: 49%, 67%) and 70% (95% C.L.: 57%, 79%) over the 3 months follow-up period, as estimated by cross-sectional community sampling, and a longitudinal sentinel house study, respectively. Given the broad confidence intervals, these values are not dissimilar to the 49% (95% C.L.: 8.2%, 71.3%) mean reduction in female *Lu. longipalpis* abundance achieved using 10mg of synthetic pheromone + insecticide in the previous CRT, which also reduced *L. infantum* transmission in dogs by 53% [18]. In that study, synthetic pheromone lures and insecticide were replaced every three months over 24–30 months until the follow-up sample.

The intervention effect in the current study appeared to persist for at least two months, but for less than three months (Figs 3 and 4). Community sample sizes at 90 days follow-up were small (3 versus 8 study blocks), however a similar temporal pattern in intervention effect was obtained from the complementary longitudinal sentinel house data. Experimental studies estimated the *Lu. longipalpis* response duration to 10mg of controlled-release synthetic pheromone was similarly about 10–12 weeks [31]. Replacement of the 10mg pheromone lure and insecticide every 3 months in the CRT was sufficient to achieve the reductions in sand flies and in canine transmission [18]. Further work is needed to extend the pheromone release rate from the dispensers towards increasing the inter-intervention interval.

Another aim of the current study was to quantify the possible spatial (spill-over) effects of the synthetic pheromone from Pheromone houses to neighbouring Untreated houses. The mean reduction in Untreated houses was 24% (95% C.L.: 0.050%, 39.8%) compared to in the True Control houses, which represented 44% (95% C.L.: 29.7%, 56.1%) of the intervention effectiveness observed in Pheromone households. Although the study was not statistically powered to detect significant intervention effects within individual study blocks, the mean estimates in the eight blocks were broadly similar (Fig 5). The variation between blocks was not related to either the distance to the nearest neighbour Pheromone houses, or to the density of Pheromone houses within 30m radius of Untreated houses. This suggests that the strength of the synthetic pheromone attraction plume was relatively constant across space, at least over the range of inter-house median distances in this sample (16m [IQR: 12.7–20.9m], range: 2.2–45.2m). This is not altogether unexpected as the distances largely fell within the known spatial attraction plume of the synthetic pheromone, estimated to be at least 30m by mark-recapture studies in this population [34].

The spatial effects were achieved by treating only 19.6% of the total houses (range: 14.4%-26.1% per study block). We did not detect a significant relationship between the variation in intervention effect (IRR) and the level of intervention coverage (Fig 6). The low coverage in this study was the result of the strict study recruitment criteria which excluded many houses. Further studies to measure the spatial effects at higher levels of coverage and at different

household densities would be informative. The study houses were a sample, whereas the true density of all houses within study blocks was higher. The median nearest neighbour distance was 6m (IQR: 4.3–8.5m; range: 0.8–42.1m), with a median of 6 houses (IQR: 4–9; range: 0–16) within a 30m radius of each house. This suggests that with greater intervention coverage, the synthetic pheromone + insecticide treatment should reduce vectors in a large proportion of the untreated community houses.

Alpha-cypermethrin SC was applied to a 2.6m$^2$ surface area of a household boundary wall or exterior outbuilding wall located away from the house. This was usually a cement-based wall (91% of treated houses). Cement is inherently porous thus the active ingredient is likely to persist on the surface for shorter periods compared to on less porous materials. For example, a comparative study of indoor substrates reported >80% 24-hr mortality of exposed *Anopheles marajoara* for <30 days when alpha-cypermethrin SC was applied to plastered or non-plastered cement surfaces, compared to 1–4 months duration when applied to acrylic painted or unpainted wood surfaces [51]. Unlike for the CRT, the insecticide concentration here was prepared at the equivalent of 80mg a.i./m$^2$ in order to achieve the target delivery concentration of 40mg a.i./m$^2$, the latter which is recommended for IRS by the Brazilian Ministry of Health [6]. The reasons for doubling the concentration was based on results of quality control studies that demonstrate substantial differences in a.i. concentrations in prepared spray tanks compared to the concentrations delivered to household walls during IRS campaigns [45,46,52,53]. To justify the approach in this study, insecticide samples from the spray tank and sprayed walls were collected systematically to test delivered and residual a.i. concentrations by HPLC currently underway.

The presence *versus* absence of household chickens influenced the intervention effectiveness, resulting in a 1.65 and 1.46 times increase in protection against female *Lu. longipalpis* numbers in Pheromone and Untreated homesteads, respectively. The previous CRT located the synthetic pheromone and insecticide at, or near to, chicken roosting sites, which is where most (60%) of the household vectors were recorded; at these sites, significant reductions in vector abundance was observed rather than inside houses or at dog sleeping sites/kennels where CDC traps were set in parallel [18]. The current study extends the coverage potential of the lure-and-kill method by demonstrating significant reductions in vector abundance in houses that did not keep animal hosts. Therefore the presence of chickens or other animal hosts are not an absolute requirement of this vector control method. Notwithstanding, the current study confirms that the *Lu. longipalpis* sex-aggregation pheromone and chicken odour is synergistic in combination, evidenced by the pre-intervention association between *Lu. longipalpis* and chicken numbers (using synthetic pheromone to capture flies); and as reported elsewhere without using synthetic pheromone [10]; and also as reflected in the dynamic processes of *Lu. longipalpis* lek formation [27–29]. Future studies could attempt to identify and synthesise host-specific kairomone attractants to be co-located with synthetic pheromone e.g. [54,55].

Animal shelters are typical aggregation sites for *Lu. longipalpis* and for more exophagic / exophilic sand fly vectors of other *Leishmania* species in the Old and New World [14, 24, 56–58]. The sex-aggregation pheromone naturally released by male *Lu. longipalpis* is suggested to help maintain a degree of site fidelity [27,28,34,44], such that there may be competition between pheromone naturally released by aggregating sand flies, and the synthetic pheromone released from lures placed elsewhere to attract the sand flies to insecticide. The precise mechanisms by which vector numbers were successfully reduced in Untreated houses could include a number of non-mutually exclusive processes: vectors are attracted away from Untreated houses by the synthetic pheromone placed in Pheromone houses; vectors are killed in sufficient numbers at Pheromone houses to generally reduce vector numbers in the nearby vicinity,

and/or to reduce emigration from Pheromone to Untreated houses. The current data do not allow us to test these alternative hypotheses, however secondary analyses suggest that the spill-over effects in Untreated houses were strongest when the Pheromone houses owned chickens, and irrespective of the presence/absence of chicken in Untreated houses (IRR = 0.52–0.78, z<-5.16, p<0.034). We also speculate that the quantities of synthetic pheromone released were sufficient to out-compete the pheromone naturally released by male flies, and sufficient to break any putative mechanism maintaining site fidelity. An enhanced attraction of *Lu. longipalpis* to larger quantities of synthetic pheromone was predicted by experimental field studies [31,44], and held true in this community study whereby 50mg compared to 10mg of synthetic pheromone attracted 4.8–6.3 times more females, and 3.9–5.4 times more males; such increases were observed in 89% of households. These quantities (10mg and 50mg) of synthetic pheromone are equivalent to natural pheromone released by 80,000 and 400,000 male *Lu. longipalpis* during nocturnal lekking periods over 3 months [31, 34].

The necessary experiments to tease apart the interactions between *Lu. longipalpis* responses to natural and synthetic pheromone, and to different host odours, have yet to be performed. The dynamics are complex and evolutionary pressures likely to differ between female and male flies: males initiate leks on or near animal hosts in the early evening, and recruit females and additional males by release of the sex-aggregation pheromone [20,27,28]. Female recruitment to leks appears to reach carrying capacity before that for males, thought to be due to a reduction in female blood-feeding success as female aggregations increase in size [20]. The reported increased emigration of females at higher densities to alternative aggregation sites may be to optimise blood-feeding success, given the association between blood-feeding success and female fecundity and survival [27].

## The implications for VL control

The spill-over of the synthetic pheromone + insecticide treatment, resulting in lower vector abundance in neighbouring Untreated households, is likely to facilitate achieving wider coverage of communities in which not all houses are accessible for treatment e.g. during IRS campaigns. Thus, reaching very high levels of community coverage may not be as critical as for other interventions [59], and we would expect programmatic deployment to achieve greater coverage than in the current study. The minimum critical coverage required for given levels of reduction in vector abundance and transmission would be useful.

The current reductions in vector abundance were achieved using only 1–2% of the insecticide required for IRS to cover a 250m$^2$ sized house, typical of endemic foci. The potential saving in insecticide cost is evident. A similar small quantity of insecticide was used to achieve the impacts against infection incidence reported in the CRT [18]. Here, the synthetic pheromone + insecticide intervention was applied once in 93 houses (5 × 10mg units per house), which reduced vector numbers in twice this number of study houses, monitored over three months. Based on WHO's guidelines of cost–effectiveness thresholds (http://www.who.int/choice/en/ accessed 5/11/20), we estimated that the Willingness-to-Pay (WTP)-defined 10mg unit cost of synthetic pheromone including the dispenser would be <$1 USD under varying implementation scenarios and levels of efficacy against canine and human infection incidence [60]. Assuming that the implementation costs are similar to programmatic IRS, the synthetic pheromone lure and insecticide components appear to be a cost-effective alternative relative to the alternative sand fly vector control methods e.g. topical insecticides for dogs [16,18], and IRS [61]; or to the substantial costs of human VL treatment [62]. The ultimate costs of this intervention method will depend on a number of factors including the intervention-interval determined by the duration of pheromone release, and residuality of the insecticide. Micro-

encapsulated lambda-cyhalothrin, or alternative formulations such as alpha-cypermethrin wettable powder (WP), Deltamethrin water dispersible granules (WG), or Pirimiphos-methyl capsule suspension (CS), which show longer durations on concrete and other materials [51,63], may be better options for future co-deployment.

Keeping chickens or other non-human hosts is not a requirement for the lure-and-kill method to reduce vectors and, by extrapolation, to protect against *L. infantum* infection and VL disease incidence. Chicken ownership is not ubiquitous and varies substantially between endemic foci (e.g. 19.7% of houses in this study; 9.3% in Araçatuba, São Paulo state; 37.1% in Marajó, Pará state, and 27.0% in Montes Claros, Minas Gerais state [64,65]). Targeting the intervention at such small fractions of the community is neither practical nor ethical. The role of chickens in zoopotentiation or zoophrophylaxis in the absence of synthetic pheromone is debated [64, 65], whereas spatial models incorporating these opposing processes indicate a net zooprophylactic outcome using the lure-and-kill method [60]. Our results suggest therefore that there may be some merit in keeping chickens to enhance the benefits of the intervention, in addition to chickens being an important household source of protein.

## Conclusions

Community deployment of the synthetic pheromone + insecticide intervention reduces the household abundance of the important sand fly vector of *L. infantum* in the Americas. The effects in treated and untreated nearby houses are encouraging, particularly in lessening the pressure to achieve high community coverage, and in requiring a much reduced quantity of insecticide compared to IRS. Supported by results of the previous CRT, an effectiveness trial against human clinical VL incidence is now warranted.

## Supporting information

**S1 Table. Numbers of households per study block sampled for *Lu. longipalpis* pre-intervention and at each follow-up round.**
(DOCX)

**S2 Table. Number of sentinel households sampled pre-intervention and at each follow-up sampling round post intervention**
(DOCX)

**S3 Table. Number of households per study block sampled pre-intervention to test synthetic pheromone attraction dose response on *Lu. longipalpis* trap numbers comparing 10mg and 50mg pheromone loaded dispensers fitted to a CDC light trap excluding the bulb.** Houses were sampled on one trap night per pheromone dose.
(DOCX)

**S4 Table. Summary of the nearest neighbour house distances (metres) between all houses within blocks 1–8.**
(DOCX)

**S5 Table. Covariates related to the logistics of intervention implementation tested for modifying effects on the intervention outcomes by inclusion in the mixed effects models.**
(DOCX)

**S1 Fig. Median numbers of male and female *Lu. longipalpis* captured at 295 households by CDC light trap fitted with 10mg and 50mg of synthetic pheromone controlled-release dispenser.** For graphical clarity data outliers are excluded.
(TIF)

**S2 Fig. Frequency distribution of the difference in number of *Lu. longipalpis* captured at 295 households by CDC light trap fitted with 50mg of synthetic pheromone a mean 53 days (95% C.L.: 51.7, 53.6; range: 21–91) after trapping using 10mg of synthetic phero-mone in the same households.** Data shown for sexes combined.
(TIF)

**S3 Fig. Median numbers of female *Lu. longipalpis* captured per trap night by CDC light trap fitted with 50mg of synthetic pheromone controlled-release dispenser in each trial arm (1 True Controls; 2 Untreated; 3 Pheromone), pre- and post-intervention.** For graphi-cal clarity data outliers are excluded.
(TIF)

**S4 Fig. Median numbers of female *Lu. longipalpis* captured per trap night by CDC light trap fitted with 50mg of synthetic pheromone controlled-release dispenser in trial arms: (A) placebo True Controls; (B) Untreated; and (C) Pheromone.** Data shown for sample rounds 1–4 represented by the median days from intervention. For graphical clarity data outli-ers are excluded.
(TIF)

**S5 Fig. Median numbers of female *Lu. longipalpis* captured per trap night by CDC light trap fitted with 50mg of synthetic pheromone controlled-release dispenser in sentinel houses in True Controls (blue bars) and Pheromone (red bars) trial arms.** Data shown for each sample round represented by the median days from intervention.
(TIF)

**S6 Fig. Frequency of the distances between Pheromone houses and their nearest-neighbour within-study block Untreated (left panel), and nearest-neighbour Pheromone house (right panel).**
(TIF)

**S7 Fig. The *Lu. longipalpis* M:F sex ratio in CDC captures pre- and post-intervention in the three treatment arms.** For graphical clarity data outliers are excluded.
(TIF)

## Acknowledgments

We thank A.C. Pinheiro at Centro de Controle de Zoonoses (CCZ) and J.E Natal at the GV health secretary office, for their local guidance. E. Dilger, K. Daag and A. Courtenay assisted with fieldwork in preparation of this study and provided useful discussions. We are indebted to the study communities for their understanding and compliance.

## Author Contributions

**Conceptualization:** James G. C. Hamilton, Orin Courtenay.

**Data curation:** Raquel Gonçalves, Cristian F. de Souza, Reila B. Rontani, Alisson Pereira, Katie B. Farnes.

**Formal analysis:** Raquel Gonçalves, Katie B. Farnes, Erin E. Gorsich, Orin Courtenay.

**Funding acquisition:** James G. C. Hamilton, Orin Courtenay.

**Investigation:** Raquel Gonçalves, Cristian F. de Souza, Reila B. Rontani, Alisson Pereira, James G. C. Hamilton.

**Methodology:** Raquel Gonçalves, Reila B. Rontani, Erin E. Gorsich, James G. C. Hamilton, Orin Courtenay.

**Project administration:** Raquel Gonçalves, Cristian F. de Souza, Reila B. Rontani, Rafaella A. Silva, Reginaldo P. Brazil, James G. C. Hamilton, Orin Courtenay.

**Resources:** Rafaella A. Silva, Reginaldo P. Brazil, James G. C. Hamilton.

**Supervision:** Erin E. Gorsich, James G. C. Hamilton, Orin Courtenay.

**Writing – original draft:** Raquel Gonçalves, Orin Courtenay.

**Writing – review & editing:** Raquel Gonçalves, Cristian F. de Souza, Reila B. Rontani, Erin E. Gorsich, Rafaella A. Silva, Reginaldo P. Brazil, James G. C. Hamilton, Orin Courtenay.

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
