## [Decision Letter · Decision Letter 0]

1 Nov 2020

Dear Dr. Courtenay,

Thank you very much for submitting your manuscript "Deployment of a synthetic pheromone of the sand fly Lutzomyia longipalpis co-located with insecticide provides community protection against the vector of Leishmania infantum." for consideration at PLOS Neglected Tropical Diseases. As with all papers reviewed by the journal, your manuscript was reviewed by members of the editorial board and by several independent reviewers. In light of the reviews (below this email), we would like to invite the resubmission of a significantly-revised version that takes into account the reviewers' comments. 

Dear Dr. Orin Courtenay

Thank you for your submission to PLoS NTDs. We have received the reviews for your manuscript and have come to a decision of "Major Revision". We suggest thoughtful consideration of the reviewers comments. In particular some of the issues raised by Reviewer 3. The reviewer raised significant concerns regarding experimental design and presentation which may require additional experimental work to address. We will be happy to consider the manuscript again after these issues have been addressed.

We cannot make any decision about publication until we have seen the revised manuscript and your response to the reviewers' comments. Your revised manuscript is also likely to be sent to reviewers for further evaluation.

Sincerely,

Paul O. Mireji, PhD

Associate Editor

Daniel Masiga

Deputy Editor

Dear Dr. Orin Courtenay

Thank you for your submission to PLoS NTDs. We have received the reviews for your manuscript and have come to a decision of "Major Revision". We suggest thoughtful consideration of the reviewers comments. In particular some of the issues raised by Reviewer 3. The reviewer raised significant concerns regarding experimental design and presentation which may require additional experimental work to address. We will be happy to consider the manuscript again after these issues have been addressed.

Reviewer's Responses to Questions

**Key Review Criteria Required for Acceptance?**

**Methods**

-Are the objectives of the study clearly articulated with a clear testable hypothesis stated?

-Is the study design appropriate to address the stated objectives?

-Is the population clearly described and appropriate for the hypothesis being tested?

-Is the sample size sufficient to ensure adequate power to address the hypothesis being tested?

-Were correct statistical analysis used to support conclusions?

-Are there concerns about ethical or regulatory requirements being met?

Reviewer #1: -Are the objectives of the study clearly articulated with a clear testable hypothesis stated? YES

-Is the study design appropriate to address the stated objectives? YES

-Is the population clearly described and appropriate for the hypothesis being tested? YES

-Is the sample size sufficient to ensure adequate power to address the hypothesis being tested? YES

-Were correct statistical analysis used to support conclusions? YES

-Are there concerns about ethical or regulatory requirements being met? YES

Reviewer #2: In a previous field study by Courtenay et al (2019) with a stratified cluster randomised trial, the effect of the synthetic pheromone co-located with a pyrethroid insecticide on the incidence of Leishmania exposure and infection in the canine reservoir, and the relative abundance of Lu. longipalpis around the households, were found to be comparable with those involving deployment of more expensive deltamethrin-impregnated Scalibor collars fitted to dogs. The present study compared the effects of the pheromone release at optimised rate one meter from the insecticide sprayed on two 2.6m outside surface areas of houses on vector numbers captured by CDC light traps with those of placebo. Vector numbers captured in nearby CDC light traps were recorded at monthly intervals over 3 months following the intervention. There was a significant reduction female vector abundance estimated by the cross-sectional and longitudinal sentinel studies. Similar reductions in male Lu. longipalpis were also observed. Interestingly, beneficial effects were also recorded at nearby placebo treated households located within the 30m range of the synthetic pheromone attraction plume. Apparently, household ownership of chickens increased the intervention effects in both treated and placebo arms, attributed to possible synergistic effect the pheromone and attractive effect of odor emitted by chicken. This is very well conducted study that has potential for downstream development.

Reviewer #3: This MS needs major revision see attached file with comments. Objectives are not clearly stated. I have several questions on the methodology too. Please see comments document attached

**Results**

-Does the analysis presented match the analysis plan?

-Are the results clearly and completely presented?

-Are the figures (Tables, Images) of sufficient quality for clarity?

Reviewer #1: -Does the analysis presented match the analysis plan? YES

-Are the results clearly and completely presented? YES

-Are the figures (Tables, Images) of sufficient quality for clarity? YES

Reviewer #2: Yes, the analyses match the plan, the results are adequately presented.

Reviewer #3: Presentation of results and interpretation needs to improve (see Comments attached)

**Conclusions**

-Are the conclusions supported by the data presented?

-Are the limitations of analysis clearly described?

-Do the authors discuss how these data can be helpful to advance our understanding of the topic under study?

-Is public health relevance addressed?

Reviewer #1: -Are the conclusions supported by the data presented? YES

-Are the limitations of analysis clearly described? YES

-Do the authors discuss how these data can be helpful to advance our understanding of the topic under study? YES

-Is public health relevance addressed?

Reviewer #2: Yes, the conclusions are clearly supported by the data.

Reviewer #3: Discussion and conclusions need to be revised (see comments attached)

**Editorial and Data Presentation Modifications?**

Reviewer #1: (No Response)

Reviewer #2: My recommendation to the authors: (i) In the Introduction, include a brief highlight of the previous study by Courtenay et al (2019, PLOS NTD) and rationale underlying the present study; and (ii) in the Discussion, include the need to identify the kairomonal blend(s) associated with chicken and dogs, and their potential for deployment with the pheromone.

Reviewer #3: Major revision is required.

**Summary and General Comments**

Reviewer #1: This is an exceptionally interesting paper showing, for the first time, reductions in South American sand flies, as vectors of human pathogens of the Leishmania genus, by co-located treatment with a synthetic sand fly aggregation pheromone and a pyrethroid insectides. This field trial follows on to look at human host related effects after an initial pioneering study, see authors’ ref 18 also by Courtenay et al., in this Journal (2019) which targeted the highly relevant dog population to leishmaniasis. Indeed this study uses households with dogs and chickens as part of the ecosystem realistically encountered and potentially valuable in further developing such control strategies as alternative attractant sources to human hosts.

The study will be of wide interest in that a pest insect pheromone is used significantly to reduce insecticide use for management of vectors of human pathogens. It is interesting also to note an apparent dose response relating to the deployment of the synthetic pheromone. This is not necessarily a normal, or even expected, observation but data here clearly suggest more pheromone - bigger effect. There is also reported sand fly reductions in the placebo controls and hypotheses are raised in relation to these observations which we hope the authors will soon test.

This work is from a highly experienced team in terms of the field trial planning and analysis of results. The work is written in an exemplary style. Consequently, in my view, this submission needs no needs no revision.

Reviewer #2: This is a good well designed and analyzed study that shows potential of the use of the pheromone at optimized rate with insecticide to control the sand fly vector of Leishmania infantum.

Reviewer #3: See Detailed comments attached (4 pages)

PLOS authors have the option to publish the peer review history of their article (what does this mean?). If published, this will include your full peer review and any attached files.

Reviewer #1: No

Reviewer #2: Yes: Ahmed Hassanali, Kenyatta University

Reviewer #3: Yes: Dr. Rajinder Kumar Saini
---

## [Decision Letter · Decision Letter 1]

17 Dec 2020

Dear Dr. Courtenay,

We are pleased to inform you that your manuscript 'Community deployment of a synthetic pheromone of the sand fly Lutzomyia longipalpis co-located with insecticide reduces vector abundance in treated and neighbouring untreated houses: implications for control of Leishmania infantum.' has been provisionally accepted for publication in PLOS Neglected Tropical Diseases.

Best regards,

Paul O. Mireji, PhD

Associate Editor

Daniel Masiga

Deputy Editor

Reviewer's Responses to Questions

**Key Review Criteria Required for Acceptance?**

**Methods**

-Are the objectives of the study clearly articulated with a clear testable hypothesis stated?

-Is the study design appropriate to address the stated objectives?

-Is the population clearly described and appropriate for the hypothesis being tested?

-Is the sample size sufficient to ensure adequate power to address the hypothesis being tested?

-Were correct statistical analysis used to support conclusions?

-Are there concerns about ethical or regulatory requirements being met?

Reviewer #1: -Are the objectives of the study clearly articulated with a clear testable hypothesis stated?

-Is the study design appropriate to address the stated objectives? Yes

-Is the population clearly described and appropriate for the hypothesis being tested? Yes

-Is the sample size sufficient to ensure adequate power to address the hypothesis being tested? Yes

-Were correct statistical analysis used to support conclusions? Yes

-Are there concerns about ethical or regulatory requirements being met? Yes

Reviewer #3: Accept revision

**Results**

-Does the analysis presented match the analysis plan?

-Are the results clearly and completely presented?

-Are the figures (Tables, Images) of sufficient quality for clarity?

Reviewer #1: -Does the analysis presented match the analysis plan? Yes

-Are the results clearly and completely presented? Yes

-Are the figures (Tables, Images) of sufficient quality for clarity? Yes

Reviewer #3: Accept revision

**Conclusions**

-Are the conclusions supported by the data presented?

-Are the limitations of analysis clearly described?

-Do the authors discuss how these data can be helpful to advance our understanding of the topic under study?

-Is public health relevance addressed?

Reviewer #1: -Are the conclusions supported by the data presented? Yes but specifically see the other reviewer to whom I refer above

-Are the limitations of analysis clearly described? Yes

-Do the authors discuss how these data can be helpful to advance our understanding of the topic under study? Yes

-Is public health relevance addressed? Yes

Reviewer #3: Authors have adequately revised the MS as per my previous comments

**Editorial and Data Presentation Modifications?**

Reviewer #1: No provided that the other reviewer to whom I refer agrees

Reviewer #3: Accept

**Summary and General Comments**

Reviewer #1: (No Response)

Reviewer #3: Authors have addressed most of the queries I had raised and the entire MS reads much better now. They have even changed the title keeping in mind my suggestion.

PLOS authors have the option to publish the peer review history of their article (what does this mean?). If published, this will include your full peer review and any attached files.

Reviewer #1: No

Reviewer #3: **Yes: **RAJINDER KUMAR SAINI

---

## [Editor Report · Acceptance letter]

29 Jan 2021

Dear Dr. Courtenay,

We are delighted to inform you that your manuscript, "Community deployment of a synthetic pheromone of the sand fly Lutzomyia longipalpis co-located with insecticide reduces vector abundance in treated and neighbouring untreated houses: implications for control of Leishmania infantum.," has been formally accepted for publication in PLOS Neglected Tropical Diseases.

Best regards,

Shaden Kamhawi

co-Editor-in-Chief

Paul Brindley

co-Editor-in-Chief
